# Host Barriers Limit Viral Spread in a Spillover Host: A Study of Deformed Wing Virus in the Bumblebee *Bombus terrestris*

**DOI:** 10.3390/v16040607

**Published:** 2024-04-15

**Authors:** Tabea Streicher, Pina Brinker, Simon Tragust, Robert J. Paxton

**Affiliations:** 1General Zoology, Institute for Biology, Martin Luther University Halle-Wittenberg, Hoher Weg 8, 06120 Halle (Saale), Germany; 2German Centre for Integrative Biodiversity Research (iDiv) Halle-Jena-Leipzig, Puschstraße 4, 04103 Leipzig, Germany

**Keywords:** RNA virus, haemolymph, faeces, shedding, *Apis mellifera*, onward transmission

## Abstract

The transmission of pathogens from reservoir to recipient host species, termed pathogen spillover, can profoundly impact plant, animal, and public health. However, why some pathogens lead to disease emergence in a novel species while others fail to establish or do not elicit disease is often poorly understood. There is strong evidence that deformed wing virus (DWV), an (+)ssRNA virus, spills over from its reservoir host, the honeybee *Apis mellifera*, into the bumblebee *Bombus terrestris*. However, the low impact of DWV on *B. terrestris* in laboratory experiments suggests host barriers to virus spread in this recipient host. To investigate potential host barriers, we followed the spread of DWV genotype B (DWV-B) through a host’s body using RT-PCR after experimental transmission to bumblebees in comparison to honeybees. Inoculation was per os, mimicking food-borne transmission, or by injection into the bee’s haemocoel, mimicking vector-based transmission. In honeybees, DWV-B was present in both honeybee faeces and haemolymph within 3 days of inoculation per os or by injection. In contrast, DWV-B was not detected in *B. terrestris* haemolymph after inoculation per os, suggesting a gut barrier that hinders DWV-B’s spread through the body of a *B. terrestris*. DWV-B was, however, detected in *B. terrestris* faeces after injection and feeding, albeit at a lower abundance than that observed for *A. mellifera*, suggesting that *B. terrestris* sheds less DWV-B than *A. mellifera* in faeces when infected. Barriers to viral spread in *B. terrestris* following oral infection may limit DWV’s impact on this spillover host and reduce its contribution to the community epidemiology of DWV.

## 1. Introduction

The transmission of a pathogen from a reservoir to a recipient host species, known as pathogen spillover, can result in the emergence of disease in the new host species [1]. Pathogen spillover is likely a regularly occurrence, often going unnoticed as it does not always lead to epidemic outbreaks [2,3,4]. With spillover events predicted to increase due to global change pressures, it is important to foster our understanding of those factors that determine whether spillover will result in disease emergence and spread in recipient host species [5].

Deformed wing virus (DWV), an (+)ssRNA virus, has become an emerging infectious disease of the honeybee *Apis mellifera* after a novel vector-based transmission route was introduced through the global invasion of the exotic, ectoparasitic mite *Varroa destructor* (henceforth: varroa) [6]. As a consequence, DWV’s prevalence and virulence, i.e., harm imposed on host honeybees, has dramatically increased over the last three decades [7,8]. Moreover, correlational studies of sympatric pollinator species have provided strong evidence for DWV spillover from *A. mellifera* into other insect species, especially bumblebees (*Bombus* spp.) [9,10,11,12,13].

Pathogen spillover requires the spatial and temporal alignment of several factors, often referred to as barriers, that affect pathogen release from the reservoir host into the environment and the probability of contact with an infectious dose by the recipient host [14]. Moreover, within-host barriers such as structural barriers, innate immune response, and molecular compatibility additionally need to be overcome by a pathogen to establish an infection and induce disease in a recipient host, potentially leading to onward transmission and disease spread [14,15,16]. 

Many bee pathogens, including viruses, enter the host body by the oral route through the ingestion of contaminated food, e.g., from shared floral resources, resulting in frequent pathogen transmission among bee species [17,18,19]. Given the ubiquity of both varroa on honeybees [20] and the DWV that they vector [21], as well as the propensity for DWV-infected honeybees to shed DWV [22], it is likely that bee species co-occurring with honeybees are frequently exposed to DWV. Whether ingestion of DWV by non-*Apis* bees can impact recipient host species and lead to onward transmission is poorly investigated, although it is of considerable importance, given the ongoing decline in insect pollinators and their important ecological and economic roles in wild and crop plant reproduction [23].

When ingested, a pathogen such as DWV reaches the digestive tract, where it might establish a primary infection of the intestinal epithelial cells [24]. One of the many functions of the insect gut is to act as a physical and chemical barrier to protect the insect cavity against pathogen spread throughout the host body [25]. This is an important modulator of disease severity experienced by the host as entering the host haemocoel gives a pathogen access to multiple tissues/organs that might additionally become infected, thus potentially increasing the burden of infection compared to a locally restricted infection of the gut epithelium. For example, when varroa carries DWV and transmits it by biting directly into the haemocoel of a pupa or adult *A. mellifera*, DWV effectively circumvents the honeybee’s gut barrier and replicates rapidly to a high titer and elevated virulence [8,26]. In a similar way, investigation of the bumblebee spillover host *Bombus terrestris* has shown that the experimental mode of transmission, i.e., oral versus vector-based transmission directly into the haemocoel, is a determinant of the disease severity of DWV infecting *B. terrestris* [27,28], as it is for *A. mellifera*, albeit with lower overall virulence in *B. terrestris*. These studies suggest that DWV is less successful at establishing an infection in *B. terrestris* when transmitted orally versus by injection, potentially due to within-host (gut) barriers that restrict DWV’s spread through the host body. 

*Bombus terrestris* host barriers to DWV spread are furthermore supported by laboratory-based transmission experiments between *A. mellifera* and *B. terrestris* [29]. When hosts were inoculated by injection into the haemocoel, infected *A. mellifera* readily transmitted DWV to both conspecifics and *B. terrestris*, whereas infected *B. terrestris* did not transmit DWV to either conspecifics or *A. mellifera* [29]. These results suggest that *B. terrestris* might shed insufficient DWV after inoculation by injection. Shedding, the release of infectious particles by an infected host, is an important determinant of the transmission potential of a host, as shedding high quantities of a pathogen facilitates the infection of new hosts and therefore pathogen spread [30]. Though *B. terrestris* is regularly infected with DWV when the virus spills over from honeybees [10,11,13], the extent to which it sheds sufficient DWV to sustain onward transmission is unclear. 

To explore (i) barriers to the impact of a virus on a non-reservoir host and (ii) the potential role of that non-reservoir host for the community epidemiology of a virus, we compared the infection dynamics of DWV in a model spillover recipient host, the bumblebee *B. terrestris*, with those in DWV’s presumed reservoir host, the honeybee *A. mellifera*. We used two routes of virus exposure: oral inoculation, mirroring faecal–oral transmission at flowers, and injection into the haemocoel, mirroring vector-based transmission, e.g., by a mite or a parasitoid or through abraded cuticle. Oral inoculation was used to investigate if and how quickly DWV develops from a primary infection of the gut into a systemic infection (measured as the presence of DWV in the haemolymph) in the reservoir host. This requires DWV to overcome the gut barrier to enter the haemocoel and would imply considerable harm to *B. terrestris*. Moreover, to investigate the lack of onward transmission observed for *B. terrestris* in laboratory experiments [29,31], we evaluated DWV shedding in faeces between *A. mellifera* and *B. terrestris* to further assess the extent to which *B. terrestris* might sustain the emergence of DWV in this bumblebee species as well as its spread through the wider community of flower-visiting insects.

## 2. Materials and Methods

### 2.1. Source of Bees

Honeybees (*Apis mellifera*) were sourced from one colony of the institute’s apiary (Halle (Saale), Germany). Bumblebees were collected directly from eight commercially sourced *B. terrestris* colonies (Koppert B.V., Berkel en Rodenrijs, The Netherlands) that were kept at 28 °C and 50–60% relative humidity. Bumblebee colonies were provided ad libitum with 50% (*w*/*v*) sucrose solution and fed with UV-irradiated defrosted organic pollen pellets (Imkerei Schachtner, Schardenberg, Austria) every other day. Both honeybee and bumblebee colonies tested negative prior to the experiment for the presence of six common honeybee viral targets (BQCV, CBPV, DWV-A, DWV-B, SBPV, SBV) using real-time PCR (henceforth: RT-PCR) (Appendix A), as described in the Supplementary Methods to [28].

### 2.2. Experimental Procedure

To investigate host barriers to within-host spread of DWV-B in *A. mellifera* and *B. terrestris*, we tracked viral dissemination from the local site of inoculation through the host body using RT-PCR following two modes of virus exposure: oral infection (inoculation with the virus by feeding individual insects), representing direct transmission that might occur on flowers or other environmental matrices, and injection (inoculation of the virus directly into the bee’s haemocoel), representing indirect transmission via a vector or through a damaged cuticle. We had four treatments (oral infection, control for oral infection, infection by injection, control for infection by injection), with a sample size of 35–40 honeybees per treatment (kept in one cage per treatment) and 21–28 bumblebees per treatment (kept in five cages per treatment). Viral spread through the host was assessed by screening the faeces and the haemolymph of experimental bees for DWV-B using one-step RT-PCR. 

To control for the honeybee host age, we used workers that had newly emerged from a frame of capped brood incubated overnight in the laboratory at 35 °C. Experimental inoculation of honeybees was performed on a single day. To control for the host age of experimental bumblebees, workers that were already present in colonies were marked by cutting the tips of their wings, enabling identification of newly emerged (unmarked) bumblebees. Groups of five to six unmarked *B. terrestris* workers were transferred to metal cages and starved of pollen for two days prior to inoculation. We thereby aimed to deplete the gut of its contents of pollen that had potentially been consumed immediately after emergence. We took this precaution because pollen can be contaminated with bee-associated viruses [32]. Inoculation of bumblebees took place on two subsequent days, and all four treatments were equally applied on both inoculation days. 

### 2.3. Preparation of Inoculum 

The DWV-B inoculum originated from the same stock used in [28] and was made from a pool of three honeybee pupae that, three days earlier, had been experimentally injected with DWV-B and subsequently found by RT-PCR to be devoid of five other common viruses (BQCV, CBPV, DWV-A, SBPV, and SBV). The final inoculum was diluted to 10^9^ genome equivalents per µL (GEs/μL) for oral inoculation or 10^7^ GEs/μL for inoculation by injection, aliquoted, and stored at −80 °C. 

Bees of control treatments were inoculated with a homogenate derived from three healthy honeybee pupae crushed in 0.5 M potassium phosphate buffer (pH 8.0) that was screened by qPCR and found to be devoid of BQCV, CBPV, DWV-A, DWV-B, SBPV, and SBV.

### 2.4. Oral Inoculation 

For feeding, the DWV-B inoculum (10^9^ GEs) and the respective control solution were mixed with 50% (*w*/*v*) sugar solution to a final volume of 5 μL. Oral inocula were coloured with FCF food dye (Carl Roth^®^, Karlsruhe, Germany) to a final concentration of 0.1% [33]. Colouring of the inoculum served as an additional control during subsequent haemolymph collection from the haemocoel to ascertain whether the gut had been accidentally punctured and the sample thus contaminated during the collection of haemolymph.

Honeybees were individually fed whilst holding them by their wings between the thumb and forefinger. Using a pipette, each bee was fed with 10^9^ GEs, or the respective control solution. After individual feeding, honeybees were transferred to one metal cage per treatment, provided with 50% v/w sugar solution ad libitum, and incubated at 30 °C for ten days.

Prior to feeding, bumblebees were deprived of food for 3–4 h to expedite their ingestion of inocula. Bumblebees were individually fed by transferring them to plastic beakers and then providing them with the viral inoculum (10^9^ GEs) (or the control solution) as described above. After successful consumption, bees were transferred back to metal cages housing five to six bumblebees per cage, provided with 50% v/w sugar solution ad libitum, and incubated at 30 °C for ten days.

### 2.5. Inoculation by Injection

Honeybees and bumblebees were immobilised on ice and subsequently injected with 10^7^ GEs in 1 μL PPB, or the respective control, between the second and third tergite using a Hamilton syringe (0.235 mm outer diameter). After injection, bees were transferred to metal cages, provided with 50% *w*/*v* sugar solution ad libitum, and incubated at 30 °C for ten days.

### 2.6. Sample Collection over Time

Samples of faeces and haemolymph were collected at three different time points (T1–T3) in the 10 days following inoculation (days post inoculation: dpi) to track the dynamics and spread of infection, as well as shedding of virus in faeces. Time point T1 was at 2 dpi for honeybees and 3 dpi for bumblebees. The difference in timing was chosen to account for the known high susceptibility of honeybees to DWV-B [34]. Time points T2 and T3 were at 6 and 10 dpi, respectively. Haemolymph was also sampled from untreated honeybees and bumblebees as an additional control to assess the infection status of experimental bees emerging from the colonies before the inoculation process (=0 dpi/T0).

### 2.7. Collection of Haemolymph Samples

The method for obtaining haemolymph largely followed [35]. Bees were immobilised on ice to reduce stress during sample collection and fixated on Styrofoam using insect pins. The abdomen was then cleaned using cotton swabs soaked in 75% ethanol. Bumblebee abdomens were carefully shaved on the spot of sample collection using a razor blade before disinfection of the abdomen. Using a binocular microscope, self-pulled pointed capillaries (micropipette 10 μL; BLAUBRAND^®^ intraMARK, Taufkirchen, Germany) were carefully inserted laterally into the integument between the second and third dorsal tergites. A few seconds after haemolymph was drawn from the bee, the capillary was carefully removed and the haemolymph sample transferred to a 0.5 mL microfuge tube, briefly centrifuged to bring haemolymph droplets to the tube’s base, placed on dry ice, and then stored individually at −80 °C. From 0.5–2 μL of haemolymph was collected per bee.

### 2.8. Collection of Faecal Samples

To obtain faecal samples, honeybees were first immobilised on ice. By applying gentle pressure on the honeybee’s abdomen, the faecal sample was directly collected into a 0.5 mL tube, briefly centrifuged, and placed on dry ice. Samples were stored at −80 °C until further analysis. Bumblebees were transferred to individual plastic beakers and observed for a maximum of 15 min until they defecated; bees that did not defecate were returned to their metal cage. Each faecal sample was transferred to an individual 0.5 mL tube using a pipette, briefly centrifuged, placed directly on dry ice, and subsequently stored at −80 °C until further analysis. The volume of collected faecal samples varied between 0.5 and 32 μL.

### 2.9. Viral RNA Extraction

Viral RNA was extracted from faecal samples by diluting each sample to 1:10 in potassium phosphate buffer (0.5 M, pH 8.0). The maximum volume of crude faeces used for RNA extraction was set to 20 μL, i.e., the maximum input volume was 200 μL of diluted faeces (Appendix A). Diluted samples were briefly vortexed and then centrifuged for 10 min at 4000× *g*. RNA was extracted from the supernatant using the High Pure Viral Nucleic Acid Kit (Roche Life Science, Penzberg, Germany) following the manufacturer’s instructions. This extraction kit contains a poly(A) carrier RNA that is added to each sample to help the elution of small quantities of RNA from the column. As recommended, ten minutes of incubation was included in this first step. Finally, RNA was eluted in 50 μL of elution buffer.

To confirm that this method was suitable for successful extraction of viral RNA from bee faeces, additional faecal samples collected from untreated bumblebees were artificially spiked with 5 × 10^7^ viral GEs, extracted simultaneously, and served as a positive control in RT-PCRs. In addition to this technical positive control for the detection of viral targets in bee faeces, we tested faecal samples from a subset of honeybees and bumblebees that had been orally inoculated with DWV-B. All these samples gave a positive signal for DWV-B according to RT-PCR (Appendix A).

Haemolymph was used directly in RT-PCRs for the detection of DWV-B, i.e., without RNA extraction, so as to enhance detection sensitivity to the virus and enable the analysis of small haemolymph sample volumes.

### 2.10. Detection of Virus by One-Step Real-Time PCR

To detect DWV in haemolymph or faecal samples, we used sensitive one-step RT-PCR with the DWV-B primers of [12] (5′-TATCTTCATTAAAACCGCCAGGCT-3′/5′-CTTCCTCATTAACTGAGTTGTTGTC-3′). For haemolymph samples, 1 μL of haemolymph was diluted 1:10 using nuclease-free water (Qiagen), of which 1 μL was used directly as the template per PCR. Analysis of faecal samples was undertaken by using 1 μL of total RNA extracted from each sample as the template per PCR reaction.

Faecal and haemolymph samples were analysed using the Luna Universal One-Step RT-qPCR kit (New England BioLabs, Ipswich, MA, USA) according to the manufacturer’s instructions in a final reaction volume of 10 μL and run on a Bio-Rad C1000 cycler. The cycling conditions followed the recommended manufacturer’s instructions and were run for 40 cycles, followed by a melt curve analysis from 50 °C to 95 °C, with 0.5 °C increments per second. The melting (dissociation) temperatures (T_m_) of the RT-PCR products were compared to the T_m_ of the positive control (78.5 °C) to confirm amplification of the correct target product, which has a size of 140 bp [12].

As additional quality measure, each sample was run in technical duplicate. Every run also included a positive control. For plates analysing haemolymph samples, we added one sample of a 1:100 diluted haemolymph sample collected from a virus-injected honeybee as a positive control. For plates analysing faecal samples, we used extracted RNA from a faecal sample which was artificially spiked with the virus, as described above (Appendix A), as a positive control. Additionally, all PCR plates contained a no template control (NTC) to validate that the reaction mix was free of contaminants.

As our aim was the molecular detection of our specific viral target (DWV-B) in collected faecal and haemolymph samples, a sample was counted as positive if one replicate gave a positive signal and the melt curve confirmed amplification of the correct target product (T_m_ = 78.5 °C). The DWV-B primers [12] were not designed to fit our specific one-step RT-PCR assay; non-specific amplification resulted in a product with a Tm of 74.0 °C when haemolymph samples were analysed. To check on non-specific amplification, a subset of PCR products was resolved by capillary electrophoresis on a QIAxcel (Qiagen, Hilden, Germany). These non-specific products had a length of 56–57 bp and are thus likely primer dimers (Appendix A).

### 2.11. Data Analysis

Figure 1, Figure 2 and Figure 3 present DWV-B Cq values as jittered dots on a box and whiskers plot showing the median (dark horizontal line), upper, and lower interquartile (shaded box) and the 1.5 × interquartile range (upper and lower whiskers) and were generated in R (version 3.6.3, R core team) with the package *ggplot2* [36]. Raw data of the experiment are given in Appendix A. To test whether DWV-B abundance varied in collected samples between the two bee species, DWV-B Cq values were compared using non-parametric comparison of means (Mann–Whitney U-test).

## 3. Results

### 3.1. Abundance of DWV-B in Host Faeces Following Oral Inoculation

DWV-B was detected in faecal samples from both bee species after oral inoculation (Figure 1). These data demonstrate that our experimental setup and method of viral detection functioned well. Moreover, they suggest that viral shedding through faeces is significantly greater for orally inoculated honeybees (mean Cq = 18.03 and 14.16 for 2 and 6 dpi, respectively; Figure 1a) than for bumblebees (mean Cq = 27.54 and 28.21 for 3 and 10 dpi, respectively; Figure 1b; Mann–Whitney U-Test: U = 0, exact *p* = 0.01; n_1_ = 6 honeybees, n_2_ = 4 bumblebees, data combined across sampling dates) despite having received the same inoculum dose (10^9^ GEs). The positive control, namely, faeces collected from an untreated bumblebee and subsequently spiked with 5 × 10^7^ DWV-B GEs, had a Cq value of 13.08, demonstrating that *B. terrestris* faeces do not inhibit the molecular detection of DWV-B.

We note that the faeces of bees fed a control solution exhibited a very low DWV-B signal (honeybees: Cq = 35.16; bumblebees: Cq = 35.68; Figure 1). This may represent a background infection of DWV-B in both bee species, possibly through the consumption of virus-contaminated pollen during larval development [32], (Appendix A).

### 3.2. Abundance of DWV-B in Host Faeces Following Inoculation by Injection

DWV-B was also detected in every sample of faeces of both honeybees and bumblebees after they had been injected in their haemocoel with DWV-B (Figure 2). At 6 dpi, differences in the dynamics of DWV-B-shedding between the two bee species became apparent. Viral shedding through faeces was at a constant low level for injected *B. terrestris* workers over the course of ten days (mean Cq = 31.33 and 32.85 for 6 and 10 dpi, respectively; Figure 2b). In contrast, DWV-B accumulated to significantly higher levels in honeybee faeces (mean Cq = 15.96 and 14.61 for 6 and 10 dpi, respectively; Figure 2a; Mann–Whitney U-Test: U = 12, *p* < 0.001; n_1_ = 11 honeybees, n_2_ = 20 bumblebees, data combined across sampling dates). This suggests that honeybees either support higher viral replication or that virus more readily passes from the haemocoel to the gut lumen and subsequently to the faeces in honeybees compared to bumblebees.

DWV-B was also detected in the vast majority of faecal samples collected from control-injected honeybees and bumblebees; it was found at very low abundance, as indicated by the high Cq values (mean Cq = 33.95 and 35.15 at 10 dpi for honeybees and bumblebees, respectively; Figure 2). These high Cq values were also seen in the faeces of honeybees and bumblebees fed a control solution (Figure 1) and support the notion of a background infection of DWV-B in our populations of bees, possibly as DWV acquired through contaminated larval food.

### 3.3. Abundance of DWV-B in Host Haemolymph Following Oral Inoculation

Feeding 10^9^ GEs of DWV-B to young honeybees quickly resulted in a systemic infection (Figure 3a). At 2 dpi, the haemolymph of two out of three honeybees tested positive for DWV-B. By 6 dpi, high viral loads were observed in all tested honeybees, indicated by low Cq values (mean Cq = 8.77; Figure 3a).

In stark contrast to honeybees, oral inoculation of *B. terrestris* workers with the same viral dose did not result in a systemic infection in any of the 19 analysed young bumblebees. Indeed, DWV-B could not be detected in haemolymph samples at any sampled time point (Figure 3b). Importantly, this is not due to our method of DWV detection as haemolymph samples collected from *B. terrestris* 1–2 h post-injection of DWV-B into the haemocoel were positive for DWV-B (T0, n = 3, mean Cq = 24.52; Figure 3b).

## 4. Discussion

In this study, we followed the infection dynamics of DWV-B in a spillover host, the bumblebee *Bombus terrestris,* compared with DWV-B’s presumed reservoir host, the honeybee *Apis mellifera,* after either oral inoculation or injection into the bee’s haemocoel to elucidate differences in host susceptibility and pathogen shedding. We found that DWV-B in its reservoir host, the honeybee, was rapidly shed in faeces after both oral inoculation and after inoculation by injection. Oral inoculation also readily led to systemic infection of honeybees, detectable as a viral transition from the gut to the haemolymph. In *B. terrestris*, oral inoculation and injection of DWV-B also led to viral shedding in faeces, albeit to a lesser extent than in honeybees. In contrast to honeybees, however, oral inoculation of bumblebees did not lead to a systemic infection, suggesting within-host barriers to the spread of DWV-B throughout the body of *B. terrestris*.

It is an interesting finding that oral inoculation through feeding a high dose of DWV-B (10^9^ GEs) to *B. terrestris* did not lead to systemic infection at a subsequent time point, indicated by the absence of the virus in host haemolymph, while the same virus dose resulted in quick virus dissemination from the gut to the haemolymph in *A. mellifera*. It suggests that DWV-B’s transit to the haemocoel of *B. terrestris* is prevented by within-host barriers such as the gut of *B. terrestris*, effectively restricting DWV-B infection to the gut epithelium.

The insect gut plays an important role as a physical and biochemical barrier, providing a first line of defence to limit pathogen access to the haemocoel [25]. It is, however, unclear which specific mechanisms act as a barrier to DWV spread within *B. terrestris*, though they clearly seem to differ from the host–virus interaction in *A. mellifera*. For comparison, Israeli acute paralysis virus (IAPV) is capable of breaching the gut barrier and establishing a systemic infection in both *B. terrestris* and *A. mellifera* following oral transmission [37]. Additionally, we found that DWV-B could reach the gut (in faeces) after injection into bumblebee haemocoel, suggesting that the gut does not represent a mere physical barrier to DWV.

Nevertheless, we cannot exclude the possibility that DWV can develop into a systemic infection in bumblebees in a context-dependent manner. For example, Cilia et al. [38] detected DWV in the heads of commercial adult *B. terrestris* workers, which suggests bees suffered a systemic infection. Furthermore, bumblebee digestive tract parasites such as *Apicystis bombi* are known to penetrate the midgut wall after host ingestion [39]. Thus, co-infection by *A. bombi* and DWV would potentially facilitate the migration of DWV to the haemocoel through a damaged gut epithelium. Alternatively, DWV might enter the bumblebee’s haemocoel via wounds of the insect cuticle [40], aided by frequent grooming behaviour that might facilitate the smearing of pathogens into such wounds. *Bombus*-specific parasitic mites or Hymenopteran parasitoids might also act as vectors for DWV by injecting virus directly into a host’s haemocoel.

The differences found between the course of DWV-B infection in *A. mellifera* versus *B. terrestris* might also be due to barriers on the molecular level, such as potential differences in the tissue tropism of DWV between the two bee species, i.e., fewer or different tissue types supporting viral replication in *B. terrestris* than in *A. mellifera*. In support of this idea, Li et al. [41] detected DWV throughout the entire body of field-collected *Bombus huntii*, though the negative strand of DWV, suggestive of viral replication, could only be detected in gut tissue. In comparison, DWV has been shown to replicate in various tissue types of *A. mellifera* [42,43,44]. Furthermore, even though *B. terrestris* is known to support replication of DWV, shown by increasing virus titres after experimental infection [27,28,29,31], the low virus shedding we observed here in infected *B. terrestris* might additionally indicate a lack of molecular compatibility between DWV and *B. terrestris*, e.g., that the needed enzymes for virus release from the host cell are insufficient. For example, avian influenza virus spilling over to humans is often characterised by a viral neuraminidase that is insufficient to cleave the viral progeny from human host proteins, thus hindering effective cell release of new virions and viral spread despite virus replication [45].

Pathogen shedding is an important determinant of the transmission potential of a host species [30] and therefore its likely contribution to the epidemiology of a multi-host pathogen [46]. We found that injection as the experimental mode of transmission of DWV-B resulted in distinct differences in viral shedding through faeces between honeybees and bumblebees. In injected honeybees, DWV-B quickly accumulated in faecal samples across the first 10 dpi, while, for injected *B. terrestris* workers, DWV-B was detected in faeces at a constant, albeit low, level post inoculation.

That *A. mellifera* and *B. terrestris* might differ in the amount of DWV excreted into the environment through faeces has already been suggested by Tehel et al. [29] to explain why, in infection experiments, *A. mellifera* could infect *B. terrestris* but not vice versa. Our results suggest that the inability of injection-inoculated *B. terrestris* to successfully transmit DWV-B onward to new hosts might be a quantitative result of generally low viral shedding. In support of this view, when inoculated orally, a likely more natural mode of infection of bumblebees in the field [9,18], we also found that *B. terrestris* shed less DWV in faeces than did *A. mellifera*. Our data on viral shedding, along with the overall lower total DWV titres accumulating in adult *B. terrestris* compared to *A. mellifera* [28,29] and the lower infection rate observed after an oral challenge [27], suggest that *B. terrestris* constitutes a less suitable host for DWV-B than *A. mellifera*. Thus, DWV’s epidemiology in multi-species communities of bees is likely driven primarily by spillover from *A. mellifera* [10,11,12], the presumed reservoir host of this virus, rather than onward transmission from infected *B. terrestris*.

Reasons for the observed differences in virus spread through the host body and virion shedding of DWV-B between the two bee species need to be further investigated to foster our understanding of the role that *B. terrestris* and other species play as alternative hosts for DWV, especially for disease spread in pollinator communities. Understanding the specific cell–virus interactions that determine DWV’s infection dynamics and the mechanisms used by DWV to release new virions from infected cells would help to elucidate underlying differences between *A. mellifera* and other bee species. Moreover, resolving the causes for the high susceptibility of *A. mellifera* compared to *B. terrestris* might provide new opportunities for veterinary interventions for disease control in honeybees.

## Figures and Tables

**Figure 1 viruses-16-00607-f001:**
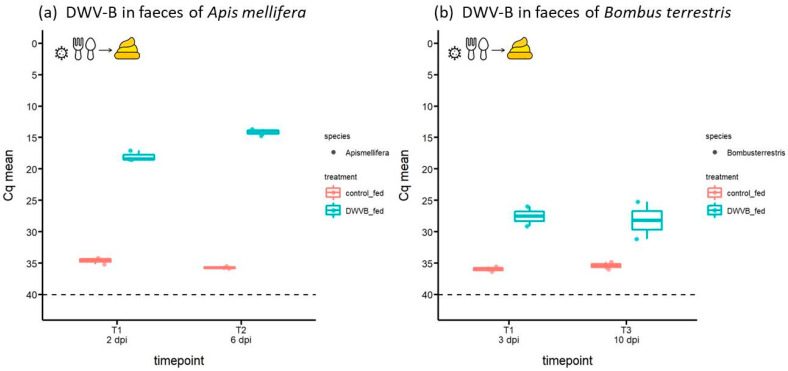
DWV-B in faeces collected from (**a**) *A. mellifera* (n = 3 per day) and (**b**) *B. terrestris* (n = 2 per day) at 2–10 days post oral inoculation (dpi) by feeding 10^9^ GEs of DWV-B or a control solution devoid of virus; Cq > 40 (below dashed line) = negative for DWV-B. The viral titre rises up the ordinate as Cq drops in value.

**Figure 2 viruses-16-00607-f002:**
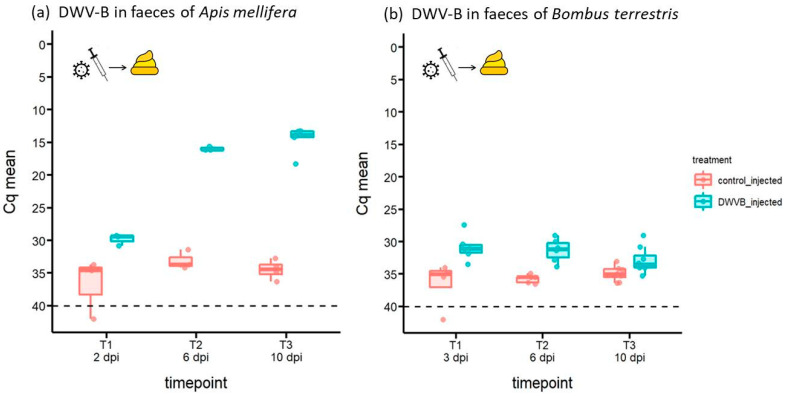
DWV-B in faeces collected from (**a**) *A. mellifera* (n = 3–5 per day) and (**b**) *B. terrestris* (n = 4–8 per day) over 2–10 days post inoculation (dpi) by injection of 10^7^ GEs DWV-B or a control solution; Cq > 40 (below dashed line) = negative for DWV-B. The viral titre rises up the ordinate as Cq drops in value.

**Figure 3 viruses-16-00607-f003:**
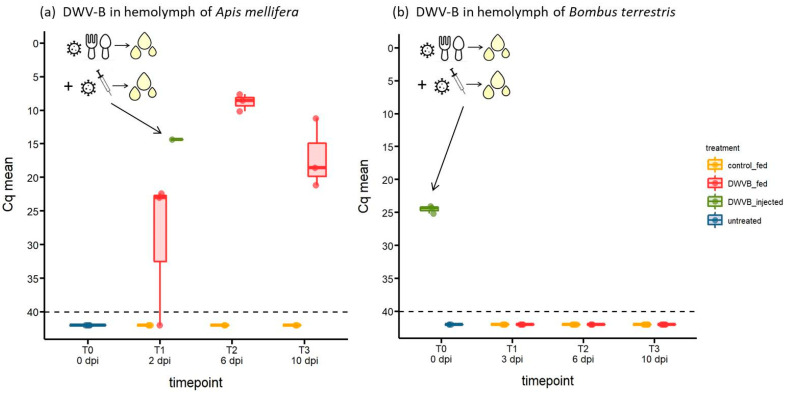
DWV-B in haemolymph collected from (**a**) *A. mellifera* (n = 3–6 per day) and (**b**) *B. terrestris* (n = 4–8 per day) at 0–10 days post inoculation (dpi) through feeding of 10^9^ GEs DWV-B or a control solution. In addition, (**a**) shows the positive control used for RT-PCT (i.e., haemolymph collected at 2 dpi from an injected honeybee; diluted 1:100), while (**b**) shows an additional control treatment for *B. terrestris* (haemolymph collected at 1–2 h post injection with 10^7^ GEs DWV-B into the haemocoel (n = 3)). Cq > 40 (below dashed line) = negative for DWV-B. The viral titre rises up the ordinate as Cq drops in value.

## Data Availability

All raw data supporting this paper are available in the Appendix A linked above.

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
