# Peer review of "Host Barriers Limit Viral Spread in a Spillover Host: A Study of Deformed Wing Virus in the Bumblebee Bombus terrestris"

_viruses, 2024, doi:10.3390/v16040607_

Round 1

Reviewer 1 Report

Comments and Suggestions for Authors

This study shows a good experiment in an important area of infection dynamics of DWV-B in a spillover host B. terrestris. The experiments were well executed, analyzed, and interpreted. The results are novel and informative and deserve to be published.

1.       Introduction

The literally review is very readable, there are 46 current references cited.

2.       Materials and methods

I have a question about the use of hemolymph directly in RT-PCR. Have you done any experiments to see if it affects the efficiency of the reaction?

 3. Results

Line 297 – 300 – low DWV-B signal was detected even in control solution of fed bees. Was this low signal detected in bees tested at the beginning of experiment (line 112)? Was PCR amplicons sequenced for verification?

Why did you use the cq value to present the data? Wouldn't a genome copy number be more appropriate? Can Cq value be affected by RNA purification?

Author Response

Reviewer(s)' Comments to Author:

Referee 1:

This study shows a good experiment in an important area of infection dynamics of DWV-B in a spillover host B. terrestris. The experiments were well executed, analyzed, and interpreted. The results are novel and informative and deserve to be published. 

#Response:

We thank the reviewer for their positive feedback, which is much appreciated. Thank you also for your comments on methodology and results, which we addressed in the following comments.

      #####

  1. Introduction 

The literally review is very readable, there are 46 current references cited. 

  1. Materials and methods 

I have a question about the use of hemolymph directly in RT-PCR. Have you done any experiments to see if it affects the efficiency of the reaction? 

#Response: 

When we tested our method for analysing haemolymph directly with RT-PCR, we tried four different concentrations (undiluted, 1:10, 1:100, 1:1000) of the same samples (n = 3) by RT-PCR. The obtained Cq values corresponded to the differences in cycle numbers that one observes for standard curves based on 1:10 dilutions (i.e. ca. 3 cycles difference between dilution steps). Though we have not directly compared the efficiency of amplification of haemolymph compared to RNA extracted and purified from the same sample, we suggest that the sensitivity of the method is probably not much affected compared to extracted and purified RNA, and may be even more sensitive. At least we can claim that our method of screening haemolymph for virus was likely equally sensitive across haemolymph samples (i.e. across Apis and Bombus) and therefore amplification efficiency per se does not affect the conclusions we reach from our data (screening of haemolymph). We acknowledge that this method was first developed by Huang et al. (2021; ref. [35] in the manuscript) as a very sensitive means of detecting virus in honey bees, who themselves did not test for the efficiency of the method.

#####

  1. Results 

Line 297 – 300 – low DWV-B signal was detected even in control solution of fed bees. Was this low signal detected in bees tested at the beginning of experiment (line 112)? Was PCR amplicons sequenced for verification? 

#Response:

At the beginning of the experiment, bumble bee colonies tested negative. We were not clear on this point so changed the text to make this explicit (change from ‘were tested’ to ‘tested negative’ at line 114).

That we nevertheless detected DWV-B in faeces from control bees during experiments might be due to the different methods used to test colonies at the start of the experiment and bees during the experiment. At the start of the experiment, we analysed RNA extracted from a pool of bees (12 bumble bees per colony for colony screening) by RNA extraction, cDNA synthesis and subsequent RT-PCR, potentially leading to insensitivity in detecting a low viral titre (we cite these methods as: ‘Supplementary Methods to [28]’). During the experiments, faecal samples were individually tested with one-step RT-PCR, known for its high sensitivity. This is an explanation for why we considered our Bombus colonies to be clean of virus at the start of the experiment, though in reality we detected DWV during the experiment. The source of that virus is, we suspect, DWV in pollen that was continuously fed to the bumble bees in their colonies. We therefore add Table S4 to the Supplementary Material, which shows low-background viral titres in pollen fed to bumble bees, and add to the ms text:

Lines 309-312: This may represent a background infection of DWV-B in both bee species, possibly through the consumption of virus-contaminated pollen during larval development [32], (Table S4; Supplementary Material).

We did not use Sanger sequencing to confirm the DWV-B in control bee faeces. However, we ran melt-curve profiles for every RT-PCR product and the melt temperatures were in accordance with the Tm of the positive control, indicating that the PCR amplicon is indeed DWV-B.

#####

Why did you use the cq value to present the data? Wouldn't a genome copy number be more appropriate? Can Cq value be affected by RNA purification? 

#Response:

We agree that Cq values are likely to be affected by RNA purification, meaning that our values from haemolymph (not purified) and faeces (purified) are not comparable. We have made sure that we do not compare them.

We also agree that using genome copy number would provide more detailed information and allow for specific statements on absolute viral titres. However, as you suggest above, RNA purity might affect the chemical reaction during PCR, thus potentially affecting the Cq value. Usually, a synthesized (purified) RNA standard would be used to generate a standard curve to allow for an absolute quantification of virus in a sample. Using a synthesized RNA standard for the calculation of virus in our non-extracted haemolymph samples would therefore presumably bring a level of uncertainty for absolute quantification for haemolymph samples. We have therefore not attempted absolute quantification. We note that Huang et al. (2021; ref. [35] in the manuscript) also did not attempt absolute quantification, possibly because generating a standard curve with RNA dilutions is far more problematic than doing so with cDNA/DNA clones.

Since we compare Cq values detected in haemolymph for both bee species, and since haemolymph from the two bee species is likely to be equally inhibitory to qPCR, we argue that our approach still allows a valid comparison between host bee species.

#####

We thank the reviewer for their insightful and constructive points of critique.

Reviewer 2 Report

Comments and Suggestions for Authors

Streicher et al here, in “Host barriers limit viral spread…”, test the potentiality of the (+)ssRNA virus Deformed Wing virus to infect the bumble bee, Bombus terrestris. This is significant as DWV has been detected in B. terrestris, not just in its reservoir host, Apis mellifera; in the context of colony collapse of the latter, and global declines in many pollinators, the ability of DWV to jump hosts and potentially establish in a new pollinator host would be damaging. Beyond that, the work intrigues at a more fundamental level – virology just doesn’t have enough examples that can and have bene studied in depth to generate general rules on novel host establishing in viruses. Therefore, Streicher et al provide compelling argument to study the potential for barriers, particularly gut, in bumble bee to DWV infection. To do this, they orally- and intrahemocoelically- (to mimic mite vector delivery of DWV, bypassing gut epithelium) introduce DWV, then assay by qPCR DWV titers in hemolymph and feces at three time points post-infection. The data indicate that DWV titers are steady in hemolymph and feces of honey bee vs bumble bee, while honey bee titers are likely 106 or more greater. The authors infer that the gut of B. terrestris likely provides a barrier to DWV systemic infection, and that there are additional barriers to establishment in B. terrestris beyond the gut (e.g., replication barriers) as injection also fails to generate high titers (thus suggesting absence both of establishment and replication and/or escape). The work is relatively simple – for example, the authors could have examined whether barriers are pre- or post-invasion, replication, and/or escape related, using a combination of gene activity relative to genome equivalents, gut tissue isolation and qPCR, microscopy, etc. – but it should provide an important springboard to do as they say and examine further the basis for these barriers, and the probability that they are sufficient to ‘prevent’ (ie reduce likelihood) species jump in the wild. Therefore, the paper is worthwhile. Additionally, both the introduction and discussion are as well composed and communicated as any I have reviewed in several years!

Minor issues: 

In the abstract, the authors highlight the issue that some viruses establish and others don’t, in new hosts. However, by statement they imply that there is no ‘grey’ area between absence (“fail to establish”) and disease incidence (“disease emergence”). This is a false binary choice, as virus infection is not always linked to disease and could in fact occur in disease absence, eventually establishing a greater potential for disease evolution in future. 

144-6: The authors note the genome equivalents used. However, what if virus produced is less infectious? It would be preferred (certainly in the future) to quantify the virus used by infectious dose rather than number of molecules. This is particularly true as the different hosts could produce virus that is more fit to reinfect the same host (or not), be more/less likely to produce defective particles, etc. I don’t believe it makes a significant difference here and thus don’t feel this experiment needs to be re-performed, but believe it is useful for future.

160-2: The authors follow inoculations by co-housing infected individuals. Would this alter probability of cycling infections, perhaps affecting infection dynamics by reducing synchrony? (Eg, by individuals ingesting fecal virus.)

191-4: Why was the tube containing hemolymph centrifuged? Was the supernatant or pellet discarded prior to storage?

297-300: I don’t understand the logic that Cq values (~35) in control bees for DWV might “represent a background infection of DWV-B”; how would this happen given the qPCR screen for 6 viruses prior to and in parallel to testing? It seems more probable to me that the primers had weak hits on the bee transcriptomes, perhaps of integrated DWV sequence. Regardless of my hypothesis (which I readily admit may be a stretch), I would like to see some empirical evidence that supports the authors’ own hypothesis such as qPCR of “contaminated larval food”, or a logical statement.

363-7: could hymenopteran parasitoids transmit the virus?

Author Response

Reviewer(s)' Comments to Author:

Referee 2:

Streicher et al here, in “Host barriers limit viral spread…”, test the potentiality of the (+)ssRNA virus Deformed Wing virus to infect the bumble bee, Bombus terrestris. This is significant as DWV has been detected in B. terrestris, not just in its reservoir host, Apis mellifera; in the context of colony collapse of the latter, and global declines in many pollinators, the ability of DWV to jump hosts and potentially establish in a new pollinator host would be damaging. Beyond that, the work intrigues at a more fundamental level – virology just doesn’t have enough examples that can and have bene studied in depth to generate general rules on novel host establishing in viruses. Therefore, Streicher et al provide compelling argument to study the potential for barriers, particularly gut, in bumble bee to DWV infection. To do this, they orally- and intrahemocoelically- (to mimic mite vector delivery of DWV, bypassing gut epithelium) introduce DWV, then assay by qPCR DWV titers in hemolymph and feces at three time points post-infection. The data indicate that DWV titers are steady in hemolymph and feces of honey bee vs bumble bee, while honey bee titers are likely 106 or more greater. The authors infer that the gut of B. terrestris likely provides a barrier to DWV systemic infection, and that there are additional barriers to establishment in B. terrestris beyond the gut (e.g., replication barriers) as injection also fails to generate high titers (thus suggesting absence both of establishment and replication and/or escape). The work is relatively simple – for example, the authors could have examined whether barriers are pre- or post-invasion, replication, and/or escape related, using a combination of gene activity relative to genome equivalents, gut tissue isolation and qPCR, microscopy, etc. – but it should provide an important springboard to do as they say and examine further the basis for these barriers, and the probability that they are sufficient to ‘prevent’ (ie reduce likelihood) species jump in the wild. Therefore, the paper is worthwhile. Additionally, both the introduction and discussion are as well composed and communicated as any I have reviewed in several years!

#Response:

We thank the referee for their complimentary and encouraging words, which are much appreciated.

#####

Minor issues: 

In the abstract, the authors highlight the issue that some viruses establish and others don’t, in new hosts. However, by statement they imply that there is no ‘grey’ area between absence (“fail to establish”) and disease incidence (“disease emergence”). This is a false binary choice, as virus infection is not always linked to disease and could in fact occur in disease absence, eventually establishing a greater potential for disease evolution in future. 

#Response:

Thank you for pointing out that this phrasing is too coarse and potentially misleading. We adjusted the sentence accordingly (‘However, why some pathogens lead to disease emergence in a novel species while others fail to establish or do not elicit disease is often poorly understood’).

#####

144-6: The authors note the genome equivalents used. However, what if virus produced is less infectious? It would be preferred (certainly in the future) to quantify the virus used by infectious dose rather than number of molecules. This is particularly true as the different hosts could produce virus that is more fit to reinfect the same host (or not), be more/less likely to produce defective particles, etc. I don’t believe it makes a significant difference here and thus don’t feel this experiment needs to be re-performed, but believe it is useful for future.

#Response:

This is an important point and we fully agree that future research needs to address this aspect by testing the viability and infectivity of shed particles in relation to host specificity/adaptation. Indeed, in another experiment that we are currently analysing, we investigated DWV’s capacity for host adaptation to B. terrestris and how this might affect viral virulence when used to re-infect B. terrestris (low) compared to A. mellifera (high). Thank you for this valuable point that we will keep to mind. Given that we used the same inoculum to infect honey bees and bumble bees, we also do not think that this point makes a difference to the experiments we performed. We believe that the current manuscript provides a helpful methodology that could be used for future experiments that aim to investigate virus dissemination and shedding.

####

160-2: The authors follow inoculations by co-housing infected individuals. Would this alter probability of cycling infections, perhaps affecting infection dynamics by reducing synchrony? (Eg, by individuals ingesting fecal virus.)

#Response:

This is a valid point; co-housing could potentially impact infection dynamics. However, we decided to co-house the bees for pragmatic reasons: their general condition can be affected in solitude, as recognised by standard methods for housing honey bees in experiments (minimum 10-20 bees per cage). A high density of conspecifics reflects the natural situation in a colony (for both honey bees and bumble bees). Indeed, honey bees kept on their own in a cage tend to die very quickly.

Honey bees defaecate little when kept in cages. To further prevent or reduce the consumption of faeces, we put filter paper on cage floors. Moreover, if co-housing actually increased the probability of infection due to contact with infected individuals, it would further support our conclusion of ‘host barriers’ in B. terrestris because, even under additional virus exposure, we observed very low shedding and limited within-host spread.

####

191-4: Why was the tube containing hemolymph centrifuged? Was the supernatant or pellet discarded prior to storage?

#Response:

A quick centrifugation step (< 10s) was included during sample collection as only small volumes of haemolymph were collected (0.5-2 µL) and we wanted to ensure that the droplet was indeed at the bottom of the tube so that the sample would freeze evenly and instantly. We did not observe pellet formation during that short centrifuge time. To make this clear, we now add to line 201:

‘briefly centrifuged to bring haemolymph droplets to the tube’s base’

####

297-300: I don’t understand the logic that Cq values (~35) in control bees for DWV might “represent a background infection of DWV-B”; how would this happen given the qPCR screen for 6 viruses prior to and in parallel to testing? It seems more probable to me that the primers had weak hits on the bee transcriptomes, perhaps of integrated DWV sequence. Regardless of my hypothesis (which I readily admit may be a stretch), I would like to see some empirical evidence that supports the authors’ own hypothesis such as qPCR of “contaminated larval food”, or a logical statement.

#Response:

Our experimental bumble bee colonies were screened for virus prior to the experiment by crushing a pool of ca. 12 adult worker bees per colony, extracting RNA, synthesising cDNA and testing by RT-PCR; all colonies tested negative by RT-PCR. Nevertheless, faeces of control bees tested positive for DWV during the experiment. This could be a result of the two different methods of either analysing a larger bee pool potentially diluting present virus below a detection threshold while faecal samples were individually tested with one-step RT-PCR, known for its high sensitivity. We argue for a low-level infection due to the circumstance that commercial bumble bee colonies are fed with honey bee pollen during colony development. Honey bee collected pollen has been shown to carry DWV and other viruses and thus might act as a source of virus. Even though methods like gamma-irradiation of pollen are used to reduce potential pathogens, gamma-irradiation is not sufficient to fully eliminate virus in pollen (Graystock et al. 2016; https://doi.org/10.1016/j.jip.2016.03.007). We used UV-light to irradiate our pollen prior to feeding to bumble bee colonies, but this likely also only limits (but not eliminates) viruses as our used pollen regularly tested positive for DWV by RT-PCR. Consumption of such pollen during development thus might cause a low level DWV infection of the gut that falls below the detection threshold when analysing RNA extracted from a pool of adult bees. Moreover, in our previous study, commercial bumble bee colonies regularly tested positive during virus screenings but always at a low level (Cq>35). We understand that this needs more elaboration; we therefore added an additional supplementary table (Table S4) that shows the results of the virus RT-PCR results for the pollen used to feed our colonies and edited the sentence in the manuscript, which reads:

(Lines 310-313): This may represent a background infection of DWV-B in both bee species, possibly through the consumption of virus-contaminated pollen during larval development [32], (Table S4; Supplementary Material).

While it is an intriguing thought that our positive results for DWV in control faeces are caused by integrated DWV snippets into the host genome, we would probably expect a different product size of the amplified ‘random’ PCR product compared to our target sequence. However, melt temperatures of PCR products in the bee control group are in accordance with the PCR positive control, suggesting that we detected non-genome integrated virus.

####

363-7: could hymenopteran parasitoids transmit the virus?

#Response:

This is a good question. In North America the parasitoid phorid fly Apocephalus borealis is known to attack bumblebees, honey bees, and paper wasps and was suggested as a potential vector for pathogens among species. Parasitic wasps are also known to attach bumble bee pupae (Alford 1975 Bumblebees. Davis-Poynter, London). We now included this possibility in the manuscript (Introduction, line 97) and furthermore mention mites and parasitoids as potential vectors known to parasitize B. terrestris in the Discussion:

(Lines 384-386) Bombus-specific parasitic mites or Hymenopteran parasitoids might also act as vectors for DWV by injecting virus directly into a host’s haemocoel.

####

 We thank the referee for their insightful and constructive points of critique.